# Population attributable risk for multimorbidity among adult women in India: Do smoking tobacco, chewing tobacco and consuming alcohol make a difference?

**Vivek K. Mishra[1], Shobhit Srivastava[2], Muhammad T.[2]\*, P. V. Murthy[3]**

**1** Department of Population Studies, Sri Venkateswara University, Tirupati, Andhra Pradesh, India,
**2** International Institute for Population Sciences, Mumbai, Maharashtra, India, **3** Department of Population Studies and Social Work, College of Arts, Sri Venkateswara University, Tirupati, Andhra Pradesh, India

\* muhammad.iips@gmail.com

**Data Availability Statement:** The study utilizes secondary source of data which is freely available in public domain through dhsprogram.com.

## Abstract

### Background

The present study aims to estimate the prevalence and correlates of multimorbidity among women aged 15–49 years in India. Additionally, the population attributable risk for multi-morbidity in reference to those women who smoke tobacco, chew tobacco, and consume alcohol is estimated.

### Methods

The data was derived from the National Family Health Survey which was conducted in 2015–16. The effective sample size for the present paper 699,686 women aged 15–49 years in India. Descriptive statistics along with bivariate analysis were used to do the preliminary analysis. Additionally, binary logistic regression analysis was used to fulfil the objectives.

### Results

About 1.6% of women had multimorbidity in India. The prevalence of multimorbidity was high among women from southern region of India. Women who smoke tobacco, chew tobacco and consume alcohol had 87% [AOR: 1.87CI: 1.65, 2.10], 18% [AOR: 1.18; CI: 1.10, 1.26] and 18% [AOR: 1.18; CI: 1.04, 1.33] significantly higher likelihood to suffer from multi-morbidity than their counterparts respectively. Population Attributable Risk for women who smoke tobacco was 1.2% (p<0.001), chew tobacco was 0.2% (p<0.001) and it was 0.2% (p<0.001) among women who consumed alcohol.

### Conclusion

The findings indicate the important role of lifestyle and behavioural factors such as smoking and chewing tobacco and consuming alcohol in the prevalence of multimorbidity among adult Indian women. The subgroups identified as at increased risk in the present study can

**Funding:** The authors received no specific funding for this work.

**Competing interests:** The authors have declared that no competing interests exist.

**Abbreviations:** NCDs, Non-Communicable Diseases; NFHS, National Family Health Survey; BMI, Body mass index; PAR, Population Attributable Risk; AOR, Adjusted Odds Ratio; CI, Confidence Interval; VIF, Variance inflation factor.

be targeted while making policies and health decisions and appropriate comorbidity management can be implemented.

# Background

Over the last couple of decades, the change in lifestyles, improvement in living conditions, and better management of communicable diseases along with improvement in medical sciences have increased the risk of non-communicable diseases in developing countries [1,2]. However, apart from non-communicable diseases, the menace of multi-morbidity is a cause of concern in both developed and developing countries. Multimorbidity is defined as the coexistence of two or more diseases in an individual [3]. The prevalence of multimorbidity was around 23% globally [4]. Another study documented that the prevalence of multimorbidity varies from 17% to over 90% in the general population [5]. Heart diseases, Asthma, Goitre or any other Thyroid disorder, Cancer, Hypertension, and diabetes, etc. are the most common non-communicable diseases that contribute to the major prevalence of multimorbidity [6].

It was found in the previous literature that the prevalence of multimorbidity increases with the increase in age [2]. As it is argued, there can be a substantial burden of multimorbidity in midlife [7]. Men and women in their midlife who suffer from multimorbidity, are more likely to suffer from various illnesses in their later years of life [7]. The quality of life of individuals also gets adversely affected due to the chaos of multimorbidity in midlife [7]. Multiple studies have suggested that multi-morbidity is more prevalent among individuals from higher socio-economic status [2,8,9]. However, some studies also argued that multimorbidity is highly prevalent among those from lower socioeconomic status [4,10,11].

Substance use is one of the most important risk factors for poor mental and physical health and subsequently the multi-morbidity especially among the vulnerable populations [12,13]. Substance use can have negative consequences on the economy and productivity of a country, and the social well-being of communities in a particular country [14]. A cross-country study showed that 4.2% of all disability-adjusted life years (DALYs) were attributed to the use of alcohol, and 1.3% of all DALYs were attributed to the use of drugs [15]. Tobacco and alcohol use disorder were found to be significantly associated with multimorbid conditions among individuals [6,16–18].

Additionally, unhealthy lifestyles like non-nutritious food may cause negative health consequences on individuals as raised blood pressure, increased blood glucose, elevated blood lipids results in overweight or obesity [6]. Obesity for instance was shown to be one of the most significant risk factors for multimorbidity among adults [5].

The reproductive age of a woman (15–49 years) [19] is considered a very important phase of her life. The burden of multimorbidity among these women may be considered devastating in terms of quality of life among them. There is a dearth of literature focusing on multimorbidity among adult women in India. Additionally, none of the literature estimated the population attributable risk for multi-morbidity in reference to those women who smoke tobacco, chew tobacco, and consume alcohol.

The present study aims to estimate the prevalence and correlates of multimorbidity among women aged 15–49 years in India. Additionally, the population attributable risk for multi-morbidity in reference to those women who smoke tobacco, chew tobacco, and consume alcohol is estimated. The study hypothesized there is a significant difference in population attributable risk for multi-morbidity in reference to those women who smoke tobacco, chew tobacco, and consume alcohol.

## Methods

### Data

The data was derived from the National Family Health Survey (NFHS-4), the fourth in the NFHS series conducted in 2015–16 It provides information on population, health, and nutrition for India and each state and union territory [20]. All four NFHS surveys have been conducted under the stewardship of the Ministry of Health and Family Welfare (MoHFW), Government of India. MoHFW designated the International Institute for Population Sciences (IIPS), Mumbai, as the nodal agency for all of the surveys. Decisions about the overall sample size required for NFHS-4 were guided by several considerations, paramount among which was the need to produce indicators at the district, state/union territory (UT), and national levels, as well as separate estimates for urban and rural areas in the 157 districts that have 30–70 percent of the population living in urban areas as per the 2011 census, with a reasonable level of precision. The NFHS-4 sample is a stratified two-stage sample [20]. The 2011 census served as the sampling frame for the selection of PSUs. PSUs were villages in rural areas and Census Enumeration Blocks (CEBs) in urban areas. PSUs with fewer than 40 households were linked to the nearest PSU. Within each rural stratum, villages were selected from the sampling frame with probability proportional to size (PPS). In each stratum, six approximately equal substrata were created by crossing three substrata, each created based on the estimated number of households in each village, with two substrata, each created based on the percentage of the population belonging to scheduled castes and scheduled tribes (SCs/STs) [20]. Four survey questionnaires (Household Questionnaire, Woman's Questionnaire, Man's Questionnaire, and Biomarker Questionnaire) were canvassed in 17 local languages using Computer Assisted Personal Interviewing (CAPI). The sample selected in NFHS survey is presented in Fig 1. The effective sample size for the present paper 699,686 women aged 15–49 years in India [20].

### Variable description

**Outcome variable.** The outcome variable was multimorbidity among men and women and coded as no and yes [3]. The variable was assessed through the question "Do you currently have any of the following diseases?". The diseases considered for measuring multimorbidity were Hypertension, Diabetes, Asthma, Goitre or any other Thyroid disorder, Heart disease and Cancer. Hypertension was measured by taking average of two systolic and diastolic reading among the respondents. Hypertension is defined as when an individual had systolic blood pressure of more than equals to 140mmHg and/or diastolic blood pressure of more than equals to 90mmHg. All other diseases were self-reported. If the respondent had two or more diseases then they were considered as multimorbid [3].

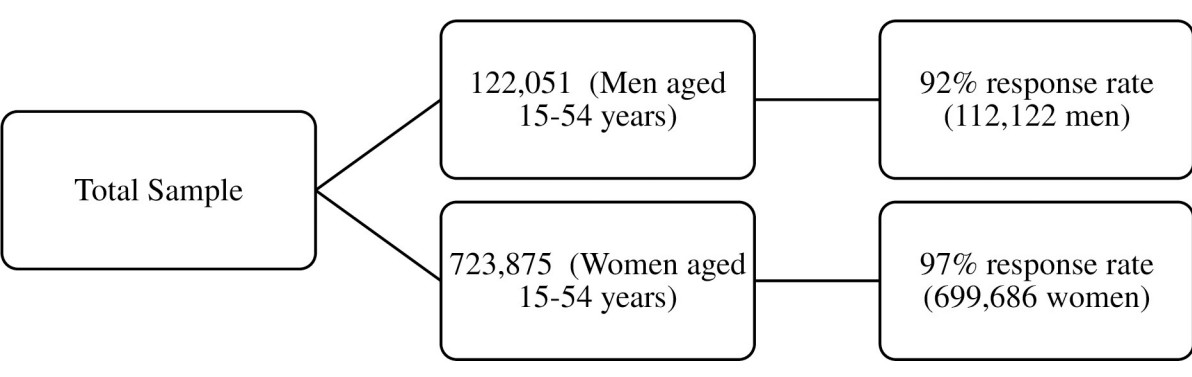

**Fig 1. Sample selection.**

**Explanatory variable.** The explanatory variables were selected based on an extensive literature review. The variables were divided into three sections that is individual characteristics, behavioural characteristics and household characteristics.

*Individual characteristics.* Age was coded as 15–24 years, 25–34 years and 35+ years. Educational status was coded as not educated, primary, secondary and higher. Working status was coded as no and yes. The variable was asked under state module hence cannot be used for multivariate analysis. Marital status was coded as never married, currently married and others. Other's included divorced/separated/widowed/deserted. Media exposure was coded as not exposed and exposed. The variable was generated using the question if the women watch television, read newspaper or listen radio. If the women was exposed all three than the response was coded as yes otherwise no. Body mass index (BMI) was recoded as underweight (less than 18.5), normal (18.5 to 24.9), overweight (25–29.9) and obese (30 and above) [21].

*Behavioural characteristics.* Cigarettes, bidis, cigars, hookah, gutkha/paan masala, paan and khaini are tobacco products consumed in India. The variable, smoking was generated using the questions a. Do you currently smoke cigarettes? b. Do you currently smoke bidis? C. Do you currently smoke cigar? and e. Do you currently smoke hookah? The variable, chewing tobacco was generated using the questions a. Do you currently chew tobacco? b. Do currently consume gutkha/paan masala with tobacco? c. Do you currently consume paan with tobacco? and e. Do currently consume khaini? Respondents who smoke cigarettes, bidis, cigars or hookah were considered a tobacco smoker. And respondents who chew tobacco in the form of gutkha/paan masala, or consume paan were considered a smokeless tobacco user [22]. Both the variables were recoded to no and yes. Women and men who consume alcohol were coded as no and yes. The variable was generated using the question "Do you drink alcohol?" [20].

*Household characteristics.* The variable wealth status was generated using the information given in the NFHS 2015–16 survey. Households were given scores based on the number and kinds of consumer goods they own, ranging from a television to a car or bicycle, and housing characteristics such as toilet facilities, source of drinking water, and flooring materials. These scores are derived using principal component analysis (PCA). National wealth quintiles are compiled by assigning the household score to each usual (de jure) household member, ranking each person in the household population by their score, and then dividing the distribution into five equal categories, each with 20 percent of the population [20]. The wealth status was coded as poorest, poorer, middle, richer and richest.

Religion was coded as Hindu, Muslim, Christian and others. Others included Buddhist, Sikh and Jain etc. [20]. Caste was coded as Scheduled Caste (SC), Scheduled Tribe (ST), Other Backward Class (OBC) and others. The Scheduled Caste include "untouchables"; a group of population that is socially segregated and financially/economically by their low status as per Hindu caste hierarchy. The SCs and STs are among the most disadvantaged and discriminated socio-economic groups in India. The OBC is the group of people who were identified as "economically, educationally and socially backward". The OBCs are considered low in the traditional caste system but are not treated as untouchables [23]. Place of residence was coded as urban and rural. Regions of India were coded as North, Central, East, North-East, West and South. Northern region included Chandigarh, Delhi, Haryana, Himachal Pradesh, Jammu & Kashmir, Punjab, Rajasthan and Uttarakhand. The central region included Chhattisgarh, Madhya Pradesh and Uttar Pradesh. The Eastern region included Bihar, Jharkhand, Odisha and West Bengal. North East region includes Arunachal Pradesh, Assam, Manipur, Meghalaya, Mizoram, Nagaland, Sikkim and Tripura. Western region includes Dadra & Nagar Haveli, Daman & Diu, Goa, Gujarat and Maharashtra. The southern region includes Andaman & Nicobar Islands, Andhra Pradesh, Karnataka, Kerala, Lakshadweep, Puducherry, Tamil Nadu and Telangana [20].

## Statistical analysis

Descriptive statistics along with bivariate analysis were used to do the preliminary analysis. Chi-square test was used to find the significance level [24]. Additionally, binary logistic regression analysis [25] was used to estimate the extent of association between multimorbidity and background factors. Variance inflation factors (VIF) were estimated to check the multicollinearity among the variables used and it was found that there was no evidence of multicollinearity [26,27]. Model-2, 3 and 4 represent the interaction effects [28,29] for age and behavioural factors on multimorbidity among women in reproductive age group in India.

Further, Population Attributable Risk (PAR) [30–32] was calculated to verify the extent of risk for multimorbidity among women who were exposed to negative behavioural factors i.e., who smoke tobacco, chew tobacco and consume alcohol [33]. The attributable risk in a population depends on the prevalence of the risk factor and the strength of its association (relative risk) with the disease [33].

## Results

### Socio-economic profile of the women aged 15–49 years in India, 2015–16

*Table 1* represents the socioeconomic profile of the women aged 15–49 years in India. It was found that 0.8% of the women consumed tobacco by smoking while 5.5% of the women consumed tobacco by chewing. Nearly 1.2% of the women consumed alcohol. About 35% of the woman belonged to 15–24 age group. Nearly 28% of the women were not educated. Around three fourth of the women reported having working status as "No" while about one-fourth of the women reported having working status as "Yes". About 73% of the women had marital status as currently married. Nearly 81% of the women reported to have no media exposure while 19% of the women reported having media exposure. Nearly 4.9% of the women were obese as per their body mass index. About one-fifth of the women were from the richest wealth quintile. Around 81% of the women belonged to the Hindu religion followed by the Muslim religion. Nearly 43% of the women were from OBC followed by SC and ST. About 65% of the women had the place of residence as rural while 35% of the women had place of residence as urban. The share of women was highest in the central region (24%) followed by the south region (23%) and east region (22%).

### Percentage of multimorbidity among women aged 15–49 years in India, 2015–16

In Fig 2 percentage of women with morbidity were represented. It was revealed that about 1.7% of women had diabetes, 1.9% had asthma, 2.2% had Goitre or any other Thyroid disorder, 1.4% has heart diseases, 0.2% had cancer, 8.7% had hypertension and about 1.6% of women had multimorbidity. *Percentage of women aged 15–49 years suffering from multimorbidity in India, 2015–16.*

Table 2 reveals the percentage of women aged 15–49 years suffering from multi-morbidity in India. The higher percentage of women from the age group 35+ years suffered from multi-morbidity. The higher percentage of women who completed primary education suffered from multi-morbidity. Working women had a higher prevalence of multi-morbidity. The prevalence of multi-morbidity was higher among women who were widowed/separated/divorced. The higher percentage of women who were exposed to media had multi-morbidity. The prevalence of multi-morbidity was high among women who were obese. The prevalence of multi-morbidity was higher among men who smoke tobacco, chew tobacco and consume alcohol.

**Table 1. Socio-economic profile of the women aged 15–49 years in India, 2015–16.**

| Background characteristics | Sample | Percentage |
|---|---|---|
| **Individual characteristics** | | |
| **Age (in years)** | | |
| 15–24 | 244518 | 35.0 |
| 25–34 | 211812 | 30.3 |
| 35+ | 243357 | 34.8 |
| **Educational status** | | |
| Not educated | 192135 | 27.5 |
| Primary | 87233 | 12.5 |
| Secondary | 331037 | 47.3 |
| Higher | 89281 | 12.8 |
| **Working status^** | | |
| No | 92996 | 76.0 |
| Yes | 29355 | 24.0 |
| **Marital status** | | |
| Never married | 159035 | 22.7 |
| Currently married | 511373 | 73.1 |
| Others | 29279 | 4.2 |
| **Media exposure** | | |
| Not exposed | 132158 | 18.9 |
| Exposed | 567528 | 81.1 |
| **Body Mass Index*** | | |
| Underweight | 153331 | 21.9 |
| Normal | 390201 | 55.8 |
| Overweight | 105038 | 15.0 |
| Obese | 34269 | 4.9 |
| **Behavioural characteristics** | | |
| **Smoke tobacco** | | |
| No | 694274 | 99.2 |
| Yes | 5412 | 0.8 |
| **Chew tobacco** | | |
| No | 661453 | 94.5 |
| Yes | 38233 | 5.5 |
| **Alcohol consumption** | | |
| No | 691048 | 98.8 |
| Yes | 8638 | 1.2 |
| **Household characteristics** | | |
| **Wealth status** | | |
| Poorest | 124054 | 17.7 |
| Poorer | 136900 | 19.6 |
| Middle | 143814 | 20.6 |
| Richer | 147978 | 21.2 |
| Richest | 146939 | 21.0 |
| **Religion** | | |
| Hindu | 563739 | 80.6 |
| Muslim | 96461 | 13.8 |
| Christian | 16620 | 2.4 |
| Others | 22866 | 3.3 |

(*Continued*)

**Table 1.** (Continued)

| Background characteristics | Sample | Percentage |
|---|---|---|
| **Caste** | | |
| Scheduled Caste | 142619 | 20.4 |
| Scheduled Tribe | 64144 | 9.2 |
| Other Backward Class | 303837 | 43.4 |
| Others | 189086 | 27.0 |
| **Place of residence** | | |
| Urban | 242225 | 34.6 |
| Rural | 457461 | 65.4 |
| **Regions** | | |
| North | 95098 | 13.6 |
| Central | 165474 | 23.7 |
| East | 154698 | 22.1 |
| North East | 24615 | 3.5 |
| West | 100535 | 14.4 |
| South | 159266 | 22.8 |
| Total | 699686 | 100.0 |

*Sample is low due to missing cases

^The question was asked on state module therefore sample is low.

## Logistic regression estimates for multimorbidity among women aged 15–49 years in India, 2015–16

Table 3 represents logistic regression estimates for multimorbidity among women aged 15–49 years in India. Women from age group 35+ years were 6.37 times significantly more likely to suffer from multi-morbidity than women from age group 15–24 years. Women who were primary educated were 10% significantly more likely to suffer from multi-morbidity than women

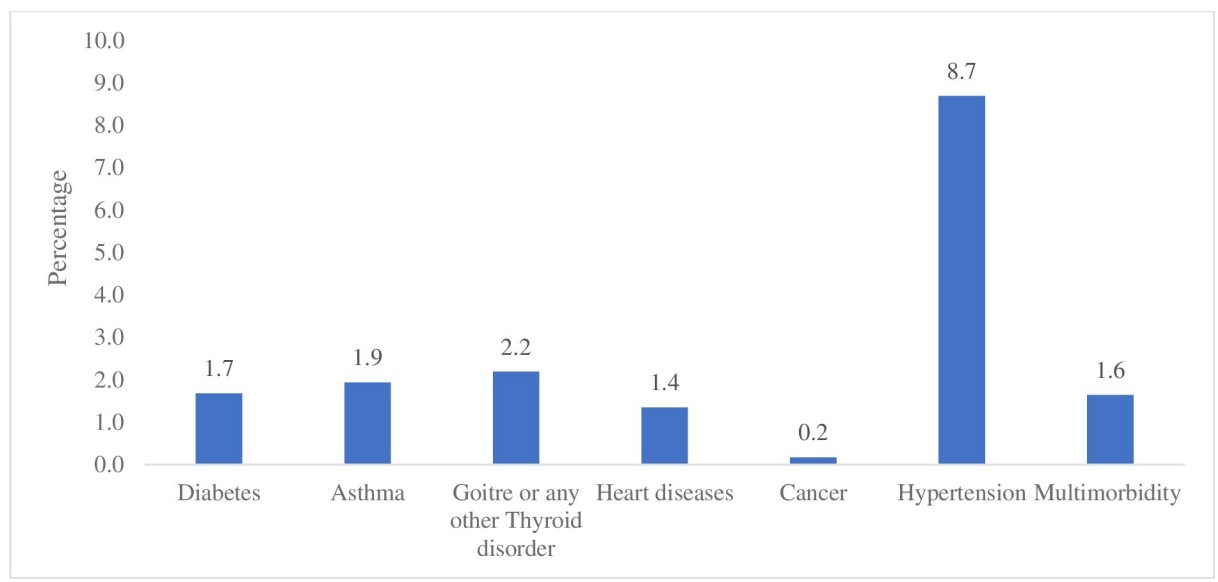

**Fig 2. Percentage of multimorbidity among women aged 15–49 years in India, 2015–16.**

**Table 2. Percentage of women aged 15–49 years suffering from multi-morbidity in India, 2015–16.**

| Background characteristics | Percentage | p-value |
|---|---|---|
| **Individual characteristics** | | |
| **Age (in years)** | | 0.001 |
| 15–24 | 0.4 | |
| 25–34 | 1.0 | |
| 35+ | 3.5 | |
| **Educational status** | | 0.001 |
| Not educated | 1.9 | |
| Primary | 2.1 | |
| Secondary | 1.5 | |
| Higher | 1.2 | |
| **Working status** | | 0.042 |
| No | 1.5 | |
| Yes | 1.6 | |
| **Marital status** | | 0.001 |
| Never married | 0.4 | |
| Currently married | 1.9 | |
| Others | 3.3 | |
| **Media exposure** | | 0.001 |
| Not exposed | 1.3 | |
| Exposed | 1.7 | |
| **Body Mass Index** | | 0.001 |
| Underweight | 0.6 | |
| Normal | 1.2 | |
| Overweight | 3.4 | |
| Obese | 6.4 | |
| **Behavioural characteristics** | | |
| **Smoke tobacco** | | 0.001 |
| No | 1.6 | |
| Yes | 4.2 | |
| **Chew tobacco** | | 0.001 |
| No | 1.6 | |
| Yes | 2.5 | |
| **Alcohol consumption** | | 0.001 |
| No | 1.6 | |
| Yes | 3.2 | |
| **Household characteristics** | | |
| **Wealth status** | | 0.001 |
| Poorest | 1.1 | |
| Poorer | 1.2 | |
| Middle | 1.5 | |
| Richer | 2.0 | |
| Richest | 2.3 | |
| **Religion** | | 0.001 |
| Hindu | 1.5 | |
| Muslim | 2.1 | |
| Christian | 2.7 | |
| Others | 1.6 | |

(*Continued*)

**Table 2.** (Continued)

| Background characteristics | Percentage | p-value |
|---|---|---|
| **Caste** | | 0.001 |
| Scheduled Caste | 1.5 | |
| Scheduled Tribe | 1.1 | |
| Other Backward Class | 1.6 | |
| Others | 2.0 | |
| **Place of residence** | | 0.001 |
| Urban | 2.1 | |
| Rural | 1.4 | |
| **Regions** | | 0.001 |
| North | 1.4 | |
| Central | 1.2 | |
| East | 1.8 | |
| North East | 1.6 | |
| West | 1.1 | |
| South | 2.4 | |
| Total | 1.6 | |

who were not educated. Women who were divorced/separated/widowed had significantly higher odds to suffer from multi-morbidity than women were never married. Women were exposed to media had 19% significantly higher odds to suffer from multi-morbidity than women who were not exposed to media. Obese women had 3.73 times significantly higher odds to suffer from multi-morbidity than women who had normal BMI. Women were smoked tobacco, chew tobacco and consume alcohol had 87%, 18% and 18% significantly higher likelihood to suffer from multi-morbidity than their counterparts respectively. Higher the wealth status higher the likelihood of women to suffer from multi-morbidity; that is, women from the richest wealth status were 51% significantly more likely to suffer from multi-morbidity than women from the poorest wealth status. Women from the Muslim religion and the Christian religion had higher odds to suffer from multi-morbidity than women from the Hindu religion. Women from the eastern region, northeastern region and southern region had significantly higher odds to suffer from multi-morbidity than women from northern region.

It was revealed in model-2 that women aged 35+ who smoke tobacco had higher odds to suffer from multimorbidity than women who is in the age group 15–24 years and do not smoke tobacco. Similarly, in model-3 it was found that women aged 35+ who chew tobacco had higher odds to suffer from multimorbidity than women who is in the age group 15–24 years and do not chew tobacco. In model-4 it was found that women aged 35+ who consume alcohol had higher odds to suffer from multimorbidity than women who is in the age group 15–24 years and do not consume alcohol.

## Population attributable risk for multimorbidity among women aged 15–49 years in India, 2015–16

*Table 4* represents s Population Attributable Risk (PAR) for multimorbidity among women who smoke tobacco, chew tobacco and consume alcohol. It was found that about 1.5% of women had multimorbidity if they smoked tobacco and 2.7% of women had multimorbidity if they do not smoke tobacco. The difference between two situations is known as Population Attributable Risk, which was measured to be 1.2% (p<0.001). Similarly, Population

**Table 3. Logistic regression estimates for multi-morbidity among women aged 15–49 years in India, 2015–16.**

| Background characteristics | Model-1 | Model-2 | Model-3 | Model-4 |
|---|---|---|---|---|
| | AOR (95% CI) | AOR (95% CI) | AOR (95% CI) | AOR (95% CI) |
| **Individual characteristics** | | | | |
| **Age (in years)** | | | | |
| 15–24 | Ref. | | | |
| 25–34 | 2.20*(2.00,2.42) | | | |
| 35+ | 6.37*(5.8,7.00) | | | |
| **Educational status** | | | | |
| Not educated | Ref. | | | |
| Primary | 1.1*(1.03,1.17) | | | |
| Secondary | 0.97(0.92,1.03) | | | |
| Higher | 0.72*(0.66,0.79) | | | |
| **Marital status** | | | | |
| Never married | Ref. | | | |
| Currently married | 1.07(0.97,1.18) | | | |
| Others | 1.27*(1.12,1.43) | | | |
| **Media exposure** | | | | |
| Exposed | Ref. | | | |
| Not exposed | 1.19*(1.12,1.28) | | | |
| **Body Mass Index** | | | | |
| Underweight | 0.75*(0.70,0.81) | | | |
| Normal | Ref. | | | |
| Overweight | 2.02*(1.93,2.13) | | | |
| Obese | 3.73*(3.52,3.96) | | | |
| **Behavioural characteristics** | | | | |
| **Smoke tobacco** | | | | |
| No | Ref. | | | |
| Yes | 1.87*(1.65,2.10) | | | |
| **Chew tobacco** | | | | |
| No | Ref. | | | |
| Yes | 1.18*(1.10,1.26) | | | |
| **Alcohol consumption** | | | | |
| No | Ref. | | | |
| Yes | 1.18*(1.04,1.33) | | | |
| **Household characteristics** | | | | |
| **Wealth status** | | | | |
| Poorest | Ref. | | | |
| Poorer | 1.12*(1.04,1.22) | | | |
| Middle | 1.14*(1.05,1.24) | | | |
| Richer | 1.32*(1.21,1.45) | | | |
| Richest | 1.51*(1.37,1.66) | | | |
| **Religion** | | | | |
| Hindu | Ref. | | | |
| Muslim | 1.48*(1.4,1.56) | | | |
| Christian | 1.28*(1.17,1.4) | | | |
| Others | 0.91(0.83,1.00) | | | |
| **Caste** | | | | |
| Scheduled Caste | Ref. | | | |

*(Continued)*

**Table 3.** (Continued)

| Background characteristics | Model-1 AOR (95% CI) | Model-2 AOR (95% CI) | Model-3 AOR (95% CI) | Model-4 AOR (95% CI) |
|---|---|---|---|---|
| Scheduled Tribe | 0.77*(0.71,0.84) | | | |
| Other Backward Class | 0.88*(0.83,0.93) | | | |
| Others | 1.12*(1.05,1.20) | | | |
| **Place of residence** | | | | |
| Urban | Ref. | | | |
| Rural | 0.99(0.95,1.04) | | | |
| **Regions** | | | | |
| North | Ref. | | | |
| Central | 1.03(0.97,1.10) | | | |
| East | 1.28*(1.19,1.37) | | | |
| North East | 1.24*(1.14,1.34) | | | |
| West | 0.70*(0.63,0.77) | | | |
| South | 1.32*(1.24,1.42) | | | |
| **Age # smoke tobacco** | | | | |
| 15–24 # No | | Ref. | | |
| 15–24# Yes | | 5.27*(3.23,8.59) | | |
| 25–34 # No | | 2.22*(2.01,2.45) | | |
| 25–34 # Yes | | 5.51*(4.04,7.5) | | |
| 35+ # No | | 6.49*(5.9,7.14) | | |
| 35+ # Yes | | 11.05*(9.4,12.99) | | |
| **Age # chew tobacco** | | | | |
| 15–24 # No | | | Ref. | |
| 15–24# Yes | | | 2.52*(1.99,3.18) | |
| 25–34 # No | | | 2.34*(2.12,2.59) | |
| 25–34 # Yes | | | 2.81*(2.38,3.32) | |
| 35+ # No | | | 6.85*(6.21,7.56) | |
| 35+ # Yes | | | 7.68*(6.84,8.63) | |
| **Age # alcohol consumption** | | | | |
| 15–24 # No | | | | Ref. |
| 15–24# Yes | | | | 3.04*(2.14,4.33) |
| 25–34 # No | | | | 2.24*(2.03,2.47) |
| 25–34 # Yes | | | | 3.55*(2.75,4.57) |
| 35+ # No | | | | 6.58*(5.98,7.24) |
| 35+ # Yes | | | | 6.51*(5.49,7.72) |

Ref: Reference

*if p<0.05; AOR: Adjusted odds ratio; CI: Confidence interval; Model-2, 3 and 4 were adjusted for individual, behavioural and household characteristics.

Attributable Risk for women who chew tobacco was 0.2% (p<0.001) and it was 0.2% (p<0.001) among women who consumed alcohol. Similarly, the PAR for women who are aged 35+ was 3.1% (p<0.001) who smoke tobacco, 1.5% (p<0.001) for who chew tobacco and 1.5% (p<0.001) for women who consume alcohol.

## Discussion

In this study using nationally representative secondary data in India, we extensively explored the prevalence of major risk factors of multimorbidity which include tobacco use, alcohol

**Table 4. Population attributable risk for multimorbidity among women aged 15–49 years in India, 2015–16.**

| Behavioural factors | Population attributable risk (PAR) | |
|---|---|---|
| | Multimorbidity | Multimorbidity among women aged 35+ years |
| **Smoke tobacco** | | |
| No | 0.015*(0.014,0.015) | 0.015*(0.014,0.015) |
| Yes | 0.027*(0.023,0.029) | 0.046*(0.041,0.052) |
| PAR | -0.012*(-0.014,-0.008) | -0.031*(-0.036,-0.026) |
| **Chew Tobacco** | | |
| No | 0.015*(0.014,0.015) | 0.015*(0.014,0.015) |
| Yes | 0.017*(0.016,0.018) | 0.029*(0.028, 0.033) |
| PAR | -0.002*(-0.003, 0.001) | -0.015*(-0.018,-0.011) |
| **Alcohol consumption** | | |
| No | 0.015*(0.014,0.015) | 0.015*(0.014,0.015) |
| Yes | 0.017*(0.015,0.019) | 0.029*(0.027, 0.034) |
| PAR | -0.002*(-0.004,-0.001) | -0.015*(-0.018,-0.012) |

CI: Confidence Interval; The analysis was controlled for individual and household characteristics.

consumption, overweight and obesity among adult women. Of the behavioural risk factors examined, smoked and smokeless tobacco use, and alcohol consumption were significantly associated with the prevalence of multimorbidity that is also observed in previous studies [34–36]. As evidence suggests, consuming alcohol has been recognized as a risk factor for many individual chronic conditions especially among women across all age groups [37]. Besides, a cross-sectional study found that women were more susceptible to the negative health effects of tobacco smoking compared to men [38].

Further, the women in the present study who had media exposure were found to be at increased risk for suffering from multimorbidity which can be explained by their increased opportunities for diving into unhealthy behaviors such as tobacco use and alcohol consumption. Besides, those women who were found to have negative anthropometric measures of overweight and obesity were also having higher likelihood of suffering from multimorbidity which is found in multiple population-based studies in India and other countries [35,39,40].

Another most consistent risk factor for multimorbidity is found to be the increasing age that can be explained by the chances of accumulating morbidities with increasing age which is similar to earlier studies showing a disadvantage of older ages along with female gender [41–43]. However, the higher prevalence of multimorbidity among women is often related to the survival bias which observes men having a shorter life expectancy than women which can be advantageous in the case of those who survive who would have better health conditions [44].

Moreover, it is documented that the burden of multi-morbidity is expected to shift from socioeconomically advantaged to disadvantaged populations in developing countries [42,45]. Consistently, we found an inverse association of education with multimorbidity with a less prevalence rate in the highly educated women. The results were in concordance with a population-based study in Brazil that found an increased risk of multimorbidity among the illiterate populations [46]. However, a positive association between household economic status which is indicated by wealth quintiles, and multimorbidity was observed which was similar to findings from low income countries showing a higher likelihood of experiencing chronic diseases among affluent communities [47]. The higher prevalence of multimorbidity among women of higher socioeconomic status in our study can be explained by increased use of health services by this group, which may increase their medical diagnoses of chronic diseases [48,49].

This study used nationally representative secondary data and the findings are generalizable for the adult women in India who are in the reproductive age of 15–49 years. However, a major drawback is that the cross-sectional study design used cannot establish causality in the associations. The lack of information on several diseases and many behavioural factors limited this study to reveal the evidence around risk factors of multimorbidity with sufficient depth. Further, we were unable to examine the impact of specific combinations of morbidities on women's health. We instead relied on a simple count of diseases as the measure of multimorbidity which warrants further investigation.

Although, the data used in our study are of 5–6 years old, given the current Covid-19 pandemic, the findings suggest the potential health challenges in future. Early observations from the current pandemic have shown that pre-existing chronic conditions, including diabetes mellitus, cardiovascular disease (CVD) and chronic lung disease predispose affected individuals to a greater risk of severe symptoms and death [50]. Also, multimorbidity in adult women may weaken their immune system and make them more susceptible to the infection and therefore, multimorbid high risk women should be given special attention in the preventive measures of current and future pandemics. Further rounds of NFHS could provide better understanding of the target population who are risk of multi-morbidity associated with socioeconomic and behavioural factors including smoking/chewing tobacco and consuming alcohol.

## Conclusion

The findings indicate the important role of lifestyle and behavioural factors such as smoking and chewing tobacco and consuming alcohol in the prevalence of multimorbidity among adult Indian women. The subgroups identified as at increased risk in the present study can be targeted while making policies and health decisions and appropriate comorbidity management can be implemented.

Since the healthcare services and related research has focused on detecting and treating single diseases among women, not much is known about the correlates of multimorbidity in poor resource settings [51]. Hence, it is important to adequately document the prevalence of comorbidities. Further, research is also required on the longitudinal effects of several combinations of morbidities on women's reproductive health that could have implications on how the resources have been distributed among vulnerable populations to bring out better health outcomes.

## Author Contributions

**Conceptualization:** Vivek K. Mishra, P. V. Murthy.

**Data curation:** Vivek K. Mishra, P. V. Murthy.

**Formal analysis:** Vivek K. Mishra.

**Investigation:** Vivek K. Mishra, P. V. Murthy.

**Methodology:** Vivek K. Mishra, P. V. Murthy.

**Supervision:** Vivek K. Mishra, Shobhit Srivastava, Muhammad T., P. V. Murthy.

**Validation:** Vivek K. Mishra, Shobhit Srivastava, Muhammad T., P. V. Murthy.

**Writing – original draft:** Vivek K. Mishra, Shobhit Srivastava, Muhammad T., P. V. Murthy.

**Writing – review & editing:** Vivek K. Mishra, Shobhit Srivastava, Muhammad T., P. V. Murthy.

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
