## [Decision Letter · Decision Letter 0]

16 Sep 2021

PONE-D-21-12186

Population attributable risk for multi morbidity among women of reproductive age in India: Do smoking tobacco, chewing tobacco and consuming alcohol make a difference?

PLOS ONE

Dear Dr. T.,

Thank you for submitting your manuscript to PLOS ONE. After careful consideration, we feel that it has merit but does not fully meet PLOS ONE’s publication criteria as it currently stands. Therefore, we invite you to submit a revised version of the manuscript that addresses the points raised during the review process.

We look forward to receiving your revised manuscript.

Kind regards,

Zila M Sanchez, PhD

Academic Editor

PLOS ONE

Additional Editor Comments (if provided):

The authors bring and important subject with data from an important survey. However, they should address several points before acceptance.

Please, consider the suggestions of the reviewers and also:

1) English editing from a native speaker. From the abstract to conclusion we can find mistakes such as "women were smoked tobacco".

2) Do not cite very specific information in the abstract (for example, the name of the states). Remember that you are writing to an international audience that don't know the state/ provinces.

3) The variables should be better describe:

- outcome: what's the questions that were used to consider a case for each disease? asthma, diabetes, etc. (do you have? do you think you have? there is a medical doctor diagnostic for this disease)? Please offer details

- explanatory - how the demographics were defined before created? for example: marital status was group in never married, married and others. What are the options that generate this categories? if you leave with someone for 10 years but is not married, in which category are you? What is included in other?

Journal Requirements:

2. We note that Figure 2 (Map-1) in your submission contain map images which may be copyrighted. All PLOS content is published under the Creative Commons Attribution License (CC BY 4.0), which means that the manuscript, images, and Supporting Information files will be freely available online, and any third party is permitted to access, download, copy, distribute, and use these materials in any way, even commercially, with proper attribution. For these reasons, we cannot publish previously copyrighted maps or satellite images created using proprietary data, such as Google software (Google Maps, Street View, and Earth). For more information, see our copyright guidelines: http://journals.plos.org/plosone/s/licenses-and-copyright.

a. You may seek permission from the original copyright holder of Figure 2 (Map-1) to publish the content specifically under the CC BY 4.0 license.  

Reviewers' comments:

Reviewer's Responses to Questions

**Comments to the Author**

1. Is the manuscript technically sound, and do the data support the conclusions?

Reviewer #1: Yes

Reviewer #2: Yes

2. Has the statistical analysis been performed appropriately and rigorously? 

Reviewer #1: Yes

Reviewer #2: Yes

3. Have the authors made all data underlying the findings in their manuscript fully available?

Reviewer #1: No

Reviewer #2: Yes

4. Is the manuscript presented in an intelligible fashion and written in standard English?

Reviewer #1: Yes

Reviewer #2: Yes

5. Review Comments to the Author

Reviewer #1: This is an interesting manuscript on multimorbidity among Indian women of reproductive age. The authors analyzed data from 699,686 women aged 15-49 years in India, from the National Family Health Survey which was conducted in 2015-16.o estudo é inovador. Segundo os autores, "none ofthe litera ture estimated the population attributable risk for multi-morbidity in reference to those women who smoke tobacco, chew tobacco, and consume alcohol. I suggest some changes and also the publication of the manuscript.

Introduction

1. I suggest the phrase "The present studyaims to estimate the prevalence and correlates of multimorbidity among women aged 15-49 years in India. Additionally, the population attributable risk for multi-morbidity in reference to those women who smoke tobacco, chew tobacco, and consume alcohol is estimated. The study hypothesized there is a significant difference in population attributable risk for multimorbidity in reference to those women who smoke tobacco, chew tobacco, and consume alcohol", be converted into a paragraph.

Methods

1. Data: I strongly suggest that the design and collection of the sample described in the sentence: "In the interviewed households, 723,875 eligible women age 15-49 were identified for individual women's interviews. Interviews were completed with 699,686 women, for a response rate of 97 percent. In all, there were 122,051 eligible men aged 15-54 in households selected for the state module. Interviews were completed with 112,122 men, for a response rate of 92 percent. The effective sample size for the present paper 699,686 women aged 15-49 years in India", whether transformed into a figure or flowchart. The transformation will facilitate the understanding of obtaining the population used.

2. Outcome variable: I didn't understand what are the "three outcome variables to fulfill the desired objective".Isn't the outcome just multimorbidity?

3.Behavioural characteristics: What are: bidis, hookah, gutkha/paan masala, paan, khaini?

4. Statistical analyses: I strongly suggest that the mathematical formulas for the presentation of the descriptive analyzes and the model be removed. Just a brief text explaining the analyzes is enough

Results

Please remove all information present in the tables from the text (% values).

Discussion:

The discussion is interesting. The authors make good arguments with the literature. However, I miss the discussion about the year of the research. Data are from 5 to 6 years ago. I suggest that the authors make a paragraph discussing possible weaknesses in the data for the current moment, considering the COVID-19 pandemic.

Reviewer #2: The subject of the manuscript is relevant, the data are very widely and detailed, and the statistical analysis is well done.

Even though the subject - multimorbity associate to behavioural characteristics (alcohol and tobacco use) - is relevant, this association is extensively reported in the scientific literature. The new/interesing brought in this paper is the gender aspect (women), the age period considered (reproductive age) and the population studied (Indian/ low income country with very specific sociocultural/behavioral charactheristics. That is exactly what the paper should better explore and address in the findings and in the discussion. It would be very relevant to better understand what the role of these specific characteristics present in this country that could be different from occidental countries where the majority of the studies have been conducted: religion, marital status, working status, place of residence (the majority of the sample live in rural area).

Furthermore, the discussion of the "women of reproductive age" was not addressed. There is no discussion/data supporting the relationship between the risk factors and the reproductive aspect of the women. Or is the term "women of reproductive age" only to describe the age group addressed by the study? If yes, my suggestion is to change the term for "adult women", and not create expectancy for data/discussion at the "reproductive" direction.

6. PLOS authors have the option to publish the peer review history of their article (what does this mean?). If published, this will include your full peer review and any attached files.

Reviewer #1: No

Reviewer #2: **Yes: **Andrea Gallassi, PhD

---

## [Author Response · Author response to Decision Letter 0]

29 Sep 2021

Additional Editor Comments:

The authors bring and important subject with data from an important survey. However, they should address several points before acceptance.

Please, consider the suggestions of the reviewers and also:

1) English editing from a native speaker. From the abstract to conclusion, we can find mistakes such as "women were smoked tobacco".

Response: The manuscript is revised by an English language expert and the grammatical and formatting errors are fixed now.

2) Do not cite very specific information in the abstract (for example, the name of the states). Remember that you are writing to an international audience that don't know the state/ provinces.

Response: Dear reviewer, I agree with your comment. The authors have now reframed the sentence to “The prevalence of multimorbidity was high among women from southern region of India”. 

3) The variables should better describe:

- outcome: what's the questions that were used to consider a case for each disease? asthma, diabetes, etc. (do you have? do you think you have? there is a medical doctor diagnostic for this disease)? Please offer details

- explanatory - how the demographics were defined before created? for example: marital status was group in never married, married and others. What are the options that generate this categories? if you leave with someone for 10 years but is not married, in which category are you? What is included in other?

Response: Dear reviewer, thank you for the in-depth comment. The authors have addressed the issue in the manuscript. Additionally, the category of marital status was recoded using the question ‘What is your current marital status?’ The responses were currently married, widowed, divorced, separated, deserted and never married. Further the variable was coded as never married, currently married and others. Others included divorced/ separated/ widowed/ deserted.

Reviewer #1: 

This is an interesting manuscript on multimorbidity among Indian women of reproductive age. The authors analyzed data from 699,686 women aged 15-49 years in India, from the National Family Health Survey which was conducted in 2015-16.o estudo é inovador. Segundo os autores, "none of the literature estimated the population attributable risk for multi-morbidity in reference to those women who smoke tobacco, chew tobacco, and consume alcohol. I suggest some changes and also the publication of the manuscript.

Introduction

1. I suggest the phrase "The present study aims to estimate the prevalence and correlates of multimorbidity among women aged 15-49 years in India. Additionally, the population attributable risk for multi-morbidity in reference to those women who smoke tobacco, chew tobacco, and consume alcohol is estimated. The study hypothesized there is a significant difference in population attributable risk for multimorbidity in reference to those women who smoke tobacco, chew tobacco, and consume alcohol", be converted into a paragraph.

Response: Thank you for the suggestion. The comment is incorporated in the revised manuscript.

Methods

1. Data: I strongly suggest that the design and collection of the sample described in the sentence: "In the interviewed households, 723,875 eligible women age 15-49 were identified for individual women's interviews. Interviews were completed with 699,686 women, for a response rate of 97 percent. In all, there were 122,051 eligible men aged 15-54 in households selected for the state module. Interviews were completed with 112,122 men, for a response rate of 92 percent. The effective sample size for the present paper 699,686 women aged 15-49 years in India", whether transformed into a figure or flowchart. The transformation will facilitate the understanding of obtaining the population used.

Response: Dear reviewer, I agree with your comment. The comment is incorporated in the manuscript. 

2. Outcome variable: I didn't understand what are the "three outcome variables to fulfill the desired objective".Isn't the outcome just multimorbidity?

Response: Dear reviewer, I am really thankful to you for this comment. Yes, the present study only has one outcome. The authors have now removed the comment. 

3.Behavioural characteristics: What are: bidis, hookah, gutkha/paan masala, paan, khaini?

Response: Cigarettes, bidis, cigars, hookah, gutkha/paan masala, paan and khaini are tobacco products consumed in India. Respondents who smoke cigarettes, bidis, cigars or hookah were considered a tobacco smoker. And respondents who chew tobacco in the form of gutkha/paan masala, or consume paan were considered a smokeless tobacco user. These are mentioned in the behavioral characteristic section now. 

4. Statistical analyses: I strongly suggest that the mathematical formulas for the presentation of the descriptive analyzes and the model be removed. Just a brief text explaining the analyzes is enough

Response: Dear reviewer, I agree with your comment. The comment is incorporated in the manuscript. 

Results

Please remove all information present in the tables from the text (% values).

Response: Dear reviewer, I agree with your comment. The authors have removed the % values from interpretation from table-2. However, in table-1, 3 and 4 authors feel that % values should be there so that the audience can easily understand the text. Please do let the authors know if any further editing can be done. 

Discussion:

The discussion is interesting. The authors make good arguments with the literature. However, I miss the discussion about the year of the research. Data are from 5 to 6 years ago. I suggest that the authors make a paragraph discussing possible weaknesses in the data for the current moment, considering the COVID-19 pandemic.

Response: Thank you so much for your observation. The possible weaknesses of the data and its relevance in the current Covid-19 pandemic have been elaborated in the discussion part of the revised version.

Reviewer #2: 

The subject of the manuscript is relevant, the data are very widely and detailed, and the statistical analysis is well done.

Even though the subject - multimorbity associate to behavioural characteristics (alcohol and tobacco use) - is relevant, this association is extensively reported in the scientific literature. The new/interesing brought in this paper is the gender aspect (women), the age period considered (reproductive age) and the population studied (Indian/ low income country with very specific sociocultural/behavioral charactheristics. That is exactly what the paper should better explore and address in the findings and in the discussion. It would be very relevant to better understand what the role of these specific characteristics present in this country that could be different from occidental countries where the majority of the studies have been conducted: religion, marital status, working status, place of residence (the majority of the sample live in rural area).

Furthermore, the discussion of the "women of reproductive age" was not addressed. There is no discussion/data supporting the relationship between the risk factors and the reproductive aspect of the women. Or is the term "women of reproductive age" only to describe the age group addressed by the study? If yes, my suggestion is to change the term for "adult women", and not create expectancy for data/discussion at the "reproductive" direction.

Response: Thank you for your suggestions. The implications of the findings are further elaborated in the revised manuscript. Yes, the term, the women of reproductive age is used to only describe the age group addressed by the study. As per your suggestion, the term has been replaced by adult women.

---

## [Editor Report · Decision Letter 1]

22 Oct 2021

Population attributable risk for multimorbidity among adult women in India: Do smoking tobacco, chewing tobacco and consuming alcohol make a difference?

PONE-D-21-12186R1

Dear Dr. T.,

We’re pleased to inform you that your manuscript has been judged scientifically suitable for publication and will be formally accepted for publication once it meets all outstanding technical requirements.

Kind regards,

Zila M Sanchez, PhD

Academic Editor

PLOS ONE
---

## [Editor Report · Acceptance letter]

26 Oct 2021

PONE-D-21-12186R1 

Population attributable risk for multimorbidity among adult women in India: Do smoking tobacco, chewing tobacco and consuming alcohol make a difference? 

Dear Dr. T.:

I'm pleased to inform you that your manuscript has been deemed suitable for publication in PLOS ONE. Congratulations! Your manuscript is now with our production department. 

Kind regards, 

on behalf of

Dr. Zila M Sanchez 

Academic Editor

PLOS ONE